# Mental Health Outcomes in Australian Healthcare and Aged-Care Workers during the Second Year of the COVID-19 Pandemic

**DOI:** 10.3390/ijerph19094951

**Published:** 2022-04-19

**Authors:** Sarah L. McGuinness, Josphin Johnson, Owen Eades, Peter A. Cameron, Andrew Forbes, Jane Fisher, Kelsey Grantham, Carol Hodgson, Peter Hunter, Jessica Kasza, Helen L. Kelsall, Maggie Kirkman, Grant Russell, Philip L. Russo, Malcolm R. Sim, Kasha P. Singh, Helen Skouteris, Karen L. Smith, Rhonda L. Stuart, Helena J. Teede, James M. Trauer, Andrew Udy, Sophia Zoungas, Karin Leder

**Affiliations:** 1School of Public Health and Preventive Medicine, Monash University, Melbourne, VIC 3800, Australia; josphin.johnson@monash.edu (J.J.); owen.eades@monash.edu (O.E.); peter.cameron@monash.edu (P.A.C.); andrew.forbes@monash.edu (A.F.); jane.fisher@monash.edu (J.F.); kelsey.grantham@monash.edu (K.G.); carol.hodgson@monash.edu (C.H.); p.hunter2@alfred.org.au (P.H.); jessica.kasza@monash.edu (J.K.); helen.kelsall@monash.edu (H.L.K.); maggie.kirkman@monash.edu (M.K.); grant.russell@monash.edu (G.R.); malcolm.sim@monash.edu (M.R.S.); helen.skouteris@monash.edu (H.S.); karen.smith@monash.edu (K.L.S.); rhonda.stuart@monashhealth.org (R.L.S.); helena.teede@monash.edu (H.J.T.); james.trauer@monash.edu (J.M.T.); andrew.udy@monash.edu (A.U.); sophia.zoungas@monash.edu (S.Z.); karin.leder@monash.edu (K.L.); 2Alfred Health, Melbourne, VIC 3004, Australia; 3School of Nursing and Midwifery, Monash University, Melbourne, VIC 3800, Australia; philip.russo@monash.edu; 4Cabrini Health, Melbourne, VIC 3144, Australia; 5Peninsula Health, Melbourne, VIC 3199, Australia; kashasingh@phcn.vic.gov.au; 6The Peter Doherty Institute for Infection and Immunity, The University of Melbourne, Melbourne, VIC 3000, Australia; 7Ambulance Victoria, Melbourne, VIC 3108, Australia; 8Monash Health, Melbourne, VIC 3168, Australia; 9Royal Melbourne Hospital, Melbourne, VIC 3050, Australia

**Keywords:** COVID-19, psychological disorders, healthcare workers, occupational health, infectious diseases epidemiology

## Abstract

Objective: the COVID-19 pandemic has incurred psychological risks for healthcare workers (HCWs). We established a Victorian HCW cohort (the Coronavirus in Victorian Healthcare and Aged-Care Workers (COVIC-HA) cohort study) to examine COVID-19 impacts on HCWs and assess organisational responses over time. Methods: mixed-methods cohort study, with baseline data collected via an online survey (7 May–18 July 2021) across four healthcare settings: ambulance, hospitals, primary care, and residential aged-care. Outcomes included self-reported symptoms of depression, anxiety, post-traumatic stress (PTS), wellbeing, burnout, and resilience, measured using validated tools. Work and home-related COVID-19 impacts and perceptions of workplace responses were also captured. Results: among 984 HCWs, symptoms of clinically significant depression, anxiety, and PTS were reported by 22.5%, 14.0%, and 20.4%, respectively, highest among paramedics and nurses. Emotional exhaustion reflecting moderate–severe burnout was reported by 65.1%. Concerns about contracting COVID-19 at work and transmitting COVID-19 were common, but 91.2% felt well-informed on workplace changes and 78.3% reported that support services were available. Conclusions: Australian HCWs employed during 2021 experienced adverse mental health outcomes, with prevalence differences observed according to occupation. Longitudinal evidence is needed to inform workplace strategies that support the physical and mental wellbeing of HCWs at organisational and state policy levels.

## 1. Introduction

The COVID-19 pandemic continues to significantly impact individuals and organisations globally. Despite Australia being one of a few countries to achieve initial control of COVID-19 [1], recurrent incursions of infections in the State of Victoria have led to an extended state of emergency (first declared 16 March 2020), with the State’s capital city of metropolitan Melbourne (population ~5 million) [2] experiencing repeated stay-at-home orders (lockdowns) throughout 2020–2021 [3]. Victoria experienced the greatest burden of COVID-19 in Australia in 2020, although the overall attack rate of 3023 per million at the end of 2020 was far less than the US (56,341/million) or the UK (33,232/million) [3,4]. An initial wave of infections (January–April 2020) was contained through public health interventions and a second wave (May–November 2020) was eventually suppressed with aggressive restrictions, including prolonged lockdowns, leading to local elimination (Figure 1). Subsequent small outbreaks were swiftly suppressed via contact tracing of identified cases and short periods of increased restrictions (including lockdowns), with case numbers remaining very low until incursion of the SARS-CoV-2 Delta variant in June 2021 led to escalating case numbers despite further lockdowns [4,5]. A shift in strategy then occurred away from elimination and towards endemicity plus high levels of vaccination [5]. The psychological burden of the pandemic upon the Australian general population has been high, and extended lockdowns in Victoria have led to increased psychological distress [6,7]. International experience has shown healthcare workers (HCWs) to be at high risk of both COVID-19 infection [8] and significant psychosocial impacts from the pandemic, including post-traumatic stress disorder (PTSD), anxiety, burnout, and depression [9,10,11,12,13]. Reports also indicate occupational disparities in COVID-19 infection risks, with HCWs, especially those in frontline roles, facing greater risks than the general community [8,14,15]. Published reports of Australian HCWs’ experiences during the pandemic’s first year (2020) suggested significant mental health impacts [16,17,18], but the effects of a second consecutive year of pandemic conditions and extensive lockdowns are unknown. Additionally, few studies have examined differences in experiences across healthcare settings or occupations.

The coronavirus in Victorian Healthcare and Aged-Care workers (COVIC-HA) cohort study has established a cohort of Australian workers from the State of Victoria across four healthcare settings to examine longitudinal impacts of the pandemic on HCWs and identify support strategies that mitigate adverse health impacts. This report presents baseline findings from the study’s first quantitative survey, describing workers’ physical and psychosocial health and perceptions of workplace responses, supported by themes arising from thematic analysis of free-text responses. These data, collected in the pandemic’s second year (2021), provide a baseline for monitoring impacts on our HCW cohort over time.

## 2. Materials and Methods

Using a mixed-methods longitudinal approach, the COVIC-HA Cohort Study is examining the physical and psychosocial health of an Australian healthcare and aged-care worker (HCW) cohort and assessing organisational responses. Organisations were recruited across four healthcare settings primarily within the south-eastern districts of Melbourne, Victoria: hospitals within the Monash Partners network, primary care practices within the Monash Practice Based Research Network, residential aged-care facilities, and Ambulance Victoria. From March 2021, participating sites’ chief executive officers, unit heads, or practice managers sent up to 5 invitation emails to all employees. Emails included an explanatory statement and web link to a brief introductory survey (Appendix A) capturing demographic and COVID-19 exposure data. Employees willing to join the longitudinal COVIC-HA cohort study provided contact details and were emailed a REDCap [19] quantitative survey link, preceded by consent forms for longitudinal data collections and (optional) data linkages and semi-structured interviews, which will be reported separately. The baseline REDCap survey (Appendix A) was open from 7 May to 18 July 2021; phone completion was also offered (Figure 1). Up to 2 email reminders were sent for non-response.

All adult employees working in any capacity at a participating study site at the time of recruitment were considered eligible. The survey was only available in English.

### Data Collection and Analysis

The baseline survey collected data on sociodemographic factors, general health, COVID-19 exposure, living situation, occupation, employment, financial circumstances, personal experiences of COVID-19, and perceptions of workplace responses. Psychological symptoms were assessed using validated instruments: anxiety (General Anxiety Disorder scale (GAD-7) [20], score ≥ 10 indicating clinically significant (moderate-to-severe) anxiety symptoms); depression (Patient Health Questionnaire (PHQ-9) [21], score ≥ 10 indicating clinically significant (moderate-to-severe) depression symptoms); post-traumatic stress (Impact of Events Scale-6 (IES-6)) [22], score ≥ 9 indicating moderate-to-severe post-traumatic stress (PTS) reaction; and burnout (abbreviated Maslach Burnout Inventory (aMBI) [23], with subdomain scores for emotional exhaustion (EE) ≥ 7, depersonalisation (DP) ≥ 4 and personal accomplishment (PA) ≤ 14 indicating moderate-to-severe burnout). Validated instruments also assessed quality of life (Personal Wellbeing Index (PWI-A) [24], range 0–100, higher scores indicating greater wellbeing) and resilience (Connor Davidson Resilience Scale 2-item (CD-RISC-2) [25], range 0–8, higher scores indicating greater resilience). Optimism about the future was assessed using a visual analogue scale (from 0, not at all optimistic, to 10, extremely optimistic) with scores of ≥8 indicating high optimism [26]. Socioeconomic status (SES) was assessed using the area level index of relative socioeconomic advantage and disadvantage (IRSAD) based on postcode of residence [27]. Participants were also invited to share experiences of the pandemic in a free text box; responses were subjected to reflexive thematic analyses according to the methods outlined by Braun and Clarke [28].

Data were analysed using STATA version 16 (StataCorp, College Station, TX, USA) and SAS version 9.4 (SAS Institute Inc., Cary, NC, USA). Participant characteristics and outcomes were summarised using means (standard deviation) or frequency (percentages). Analyses compared outcomes between five different occupations (doctors, nurses, allied health, paramedics, and ‘other’ (mainly administrative or support staff)). A prespecified set of variables were included in regression models to address potential confounders: age, gender, SES, and an indicator for date of participation. A linear term for age was included; SES decile was included as a categorical predictor in models for continuous outcomes, and as a continuous predictor for binary outcomes. For continuous outcomes, linear mixed models were fitted, including random intercepts for work site (with results expressed as adjusted difference in mean outcome (with 95% CIs) between work professions, using paramedics as the reference category. An F-test with 4 degrees of freedom was used to compare outcomes across the five occupations. For binary outcomes, random effects models were fitted using a logarithmic link function to estimate adjusted risk ratios and 95% CIs directly. The Kenward–Roger small sample correction was employed for continuous and binary outcomes [29]. Occasional convergence problems encountered were managed by removing the least important adjustment variable until convergence was obtained.

This study was approved through the Victorian Streamlined Ethical Review Process (SERP: Project 68086) and registered with ANZCTR (ACTRN12621000533897). Reporting followed the strengthening the reporting of observational studies in epidemiology (STROBE) guidelines checklist (Appendix A) [30].

## 3. Results

There were 4592 responses to the introductory survey, with 40% of respondents (n = 1823) expressing interest in longitudinal study participation and 984 eligible participants completing the baseline survey (Appendix A). Almost three-quarters were female, and half were <45 years of age. Participants were from different healthcare settings and occupations, almost one-third being nurses (Table 1). One in five (20.7%) reported being furloughed and 2.2% reported a COVID-19 infection.

### 3.1. Mental Health and Wellbeing

Overall, 22.5% of participants reported moderate–severe depression, 14.0% moderate–severe anxiety, and 20.4% moderate–severe PTS symptoms (Table 2). Thoughts of being better off dead or self-harm (PHQ-9, item 9) on at least some days were reported by 8.8%; 60.9% reported being irritable (GAD-7, item 6) on at least some days. Between-group differences in mean scores and proportions with moderate–severe symptoms were observed in adjusted analyses, with nurses and paramedics generally reporting higher prevalence of psychological symptoms than doctors and allied health professionals (Table 2 and Table 3). Paramedics reported higher prevalence of moderate-to-severe burnout in the depersonalisation and personal accomplishment subdomains. Differences were also observed in subjective wellbeing, overall life satisfaction scores, and optimism scores between occupations, with scores among nurses and paramedics generally lower than doctors and allied health professionals (Table 2 and Table 3). High levels of resilience were observed across all occupations.

### 3.2. Work Home and Lives

Concerns about contracting COVID-19 at work (62.5% of participants) and transmitting COVID-19 (71.4%) were common, with paramedics more likely to report these concerns than those from other occupational groups (Appendix A). Almost half of all respondents were worried about their vulnerability to serious complications (48.3%), highest amongst paramedics and nurses. A third of respondents overall (32.4%) and almost half of the nurses had considered leaving their profession since the start of the pandemic. Approximately one-quarter of the cohort reported redeployment (27.1%), increased paid working hours (26.1%), decreased household incomes (25.7%) and income concerns (22.3%), and more than a third (39.8%) reported that home responsibilities had interfered with work. Almost 40% experienced social isolation, with family and friends avoiding contact given their ‘high-risk’ work, the highest among paramedics and nurses (Appendix A). Increased alcohol consumption was reported by almost one-third of respondents. Mental health support had been sought by 35.7% overall, and by almost half the paramedics.

### 3.3. Perspectives on Vaccination and Workplace Supports

Over 90% of respondents felt their organisations had kept them well informed on workplace changes and most reported availability of support services at work (78.3%) and easy access to COVID-19 testing (81.3%), although paramedics reported fewer affirmative responses to these questions (Appendix A). Almost two-thirds of respondents reported a high degree of confidence that the personal protective equipment (PPE) currently available to them was adequate to protect them when managing patients with confirmed or suspected COVID-19; confidence was lower in paramedics as compared to other occupational groups. More than half of respondents felt comfortable to voice concerns (62.0%) and felt that their concerns were responded to by their organisation (56.5%). Over 90% of respondents reported receiving at least one COVID-19 vaccine dose and agreed that all HCWs should be vaccinated.

### 3.4. HCW Experiences

Forty-four percent (441 participants; Appendix A) provided a free-text response relating to their pandemic experience. Analysis identified four dominant themes (Table 4). Working in a complex and changing environment was described as physically and emotionally draining; workers cited frequent policy and practice changes, a greater need to provide emotional support to patients and residents due to visitor restrictions, and difficulty “switching off” as contributing factors. COVID-19 restrictions were considered to negatively impact patient/resident care, with concerns about long-term patient health impacts voiced. Fear of bringing COVID-19 home to loved ones was prominent, driving some to minimise interactions. Professional isolation was also reported, especially among those working from home and/or primarily via Telehealth. Positive experiences included a sense of pride in providing empathetic and respectful care in challenging times and renewed perspectives on what was important in life.

## 4. Discussion

The COVID-19 pandemic has significantly affected the psychosocial health and wellbeing of HCWs [8,9,10,11,12,13,14,15,16,17,18]. Our study builds on existing literature by reporting data from almost 1000 Australian HCWs from the State of Victoria during the pandemic’s second year and comparing outcomes according to occupation. Prevalence of clinically significant symptoms of depression, anxiety, PTS symptoms, and burnout in our cohort are concerning, but appear lower than those reported by similar studies performed during the pandemic’s first year. For example, a 2020 Australian HCW study reported similar levels of clinically significant depression symptoms (28%), but higher levels of burnout and almost two-fold greater prevalence of anxiety (28%) and PTS (40.5%) symptoms [17]. International surveys among HCWs have reported even higher levels of depression, anxiety and PTS symptoms (up to 45%, 40%, and 49%, respectively), as have community-based Australian surveys (clinically significant depression and anxiety up to 46% and 41%, respectively) (Appendix A). The average wellbeing score (69.7) among our cohort was lower than in a 2020 Victorian community survey (75.8) [31], but higher optimism was reported compared to a 2020 study of adult Australians [26].

Our study is the only Australian and one of few reported studies assessing COVID-19 impacts on HCWs in 2021. Comparisons across studies are complicated by differences in measurement instruments and recruitment strategies (geographical coverage, HCW versus community samples, included occupational groups), but the timing of our survey may also have contributed to lower prevalence of psychological distress than previously described. Although the cumulative effects of prolonged work stress, isolation, and lockdown fatigue would presumably worsen mental health over time, factors related to the setting and timing of our survey rollout may have counterbalanced these. In the months preceding and during our survey rollout, the State of Victoria experienced very low case numbers of COVID-19, paired with greater PPE accessibility (compared with earlier pandemic stages), increasing vaccination rates, and fewer restrictions in place (Figure 1). Additionally, opportunities for instigation of organisational mitigation measures have increased over time, and we found that most respondents felt well-informed on workplace changes and agreed that support services were available. Institutional responses have previously been shown to affect psychological health [32], with reports from current and prior pandemics suggesting that HCWs value clear communication, prioritisation of staff safety, and shared decision-making [33]. Furthermore, while HCWs face greater risks of COVID-19 exposure and encountering distressing patient care situations, other factors may have moderated impacts; for example, people in the workforce generally have better mental health than those who are not working [34], and stable employment meant relatively few of our cohort reported household financial concerns. HCWs also typically have greater access to up-to-date and reliable information and a propensity to feel a sense of pride and purpose in their work when rising to challenges. These factors make it difficult to disentangle psychological impacts of the ongoing pandemic specific to HCWs.

Despite relatively little community transmission in the State of Victoria prior to survey roll-out, almost one in four study participants had experienced a COVID-19 infection or period of furlough, and approximately two-thirds were worried about contracting COVID-19 and transmitting infections to others. Perceived social isolation due to high-risk work was reported by 40% of our cohort, and fear of bringing COVID-19 home to friends and family was also reflected in free-text responses. In an early 2020 study of Australian and New Zealand paediatricians, higher proportions were concerned about acquiring infection (86%) and transmitting COVID-19 (92%) [35], but corresponding rates of concern in a community setting were lower (26% and 53%, respectively) [36]. Increased alcohol use was reported by almost a third of respondents, comparable to other Australian HCWs [37] and community-based studies (21–35%, Appendix A).

A key finding in our cohort was that paramedics and nurses reported poorer mental health and higher levels of social isolation than doctors or allied health staff. Comparative data across occupational groups from Australian pre-pandemic surveys is lacking as previous investigations have focused on a single occupational group. However, pre-pandemic Australian data show that nurses and paramedics experience higher rates of adverse mental health outcomes compared to the general population [38,39] and international data show higher incidence rates of work-related mental ill health in ambulance staff and nurses compared to doctors and teachers [40]. Our observed between-occupation differences in psychological distress could relate to paramedic and nurse positions at the forefront of the pandemic response in working conditions that often predispose to greater exposure risks. Nurses frequently have prolonged patient-contact times requiring long periods in PPE, and paramedics may confront volatile emergency situations requiring rapid PPE donning and doffing. Other contributing factors may include shift work, employment grade, job control, workplace safety, and involvement in decision-making [41,42,43]. More nurses and paramedics reported seeking mental health support pre-pandemic (51% versus 45.6% overall). Paramedics also reported the highest rates of seeking mental health support since the start of the pandemic (46%), possibly attributable to their lower self-reported perceptions of organisational responsiveness and comfort in voicing concerns, or potentially reflecting the high availability of mental health supports within their organisation. Together with the paucity of pre-pandemic Australian studies comparing mental health status across HCW occupations, the specific role the pandemic has played in our observed occupational variability in outcomes is uncertain.

Our study has several limitations. First, we included only select healthcare organisations readily accessible for recruitment by investigators, thereby limiting broader generalisability of our findings. Over 80% of our cohort worked in large public or private hospital settings where infection control, testing, and other supports may be more readily available. Despite dedicated attempts to achieve greater aged-care recruitment, we were unsuccessful at engaging additional sites, likely reflecting the disproportionate burden of COVID-19 on this sector and the consequent lack of resources to engage in research. Small numbers of participants in aged care and primary care settings precluded meaningful analyses across healthcare settings. Second, although we used validated instruments to measure mental health outcomes [20,21,22,23,24,25], self-reported measures may not directly correlate with clinical diagnostic interviews [44]. However, as our primary outcome measures used validated scales and we analysed each outcome measure separately, we do not expect our results to have been impacted by common method variance bias. Third, convenience sampling, voluntary participation, and low response rates likely resulted in response biases. Attempts to minimise bias included addressing potential confounders by adjusting for age, gender, socioeconomic status, and date of participation in analyses. Notably, although our cohort was only composed of 21% of the 4592 completing the introductory survey, the demographic profiles of these two groups were nearly identical (Appendix A). An estimated 50,000 study invitations were emailed to workers across the different settings, but recruitment messages were often included within broader institutional newsletters (limiting visibility), therefore the exact response rates and effects of responder bias are uncertain. Compared to estimates of all participating organisations’ employees, our cohort was slightly older (63% < 45 years versus 51%) and included slightly more females (73% versus 62%), but the proportion who were neither furloughed nor infected with COVID-19 were similar (78% versus 81%). Finally, our cohort was composed of HCWs who were employed throughout the pandemic, thereby potentially subject to biases related to employment stability and healthy-worker effects [34]. Nevertheless, our data provide an indication of the mental health impact of COVID-19 on HCWs across a range of occupations in the pandemic’s second year, and ongoing monitoring of this longitudinal cohort will enable us to monitor trends over time.

## 5. Conclusions

In conclusion, we found a concerning frequency of clinically significant depression, anxiety, and PTS symptoms among Australian HCWs in the second year of the COVID-19 pandemic. While comparisons with earlier studies are problematic, our findings do not suggest marked worsening of psychological distress as the pandemic has progressed, potentially related to the relatively low COVID-19 case numbers during and prior to the survey period, high levels of HCW resilience, high vaccination rates, and generally positive perceptions of organisational communication and support services. However, differences between occupational groups were seen and burnout levels were high. Given the ongoing risk of COVID-19 and evidence of significant impacts on HCWs, it is essential that policymakers and healthcare organisations instigate sustainable mitigation strategies and targeted interventions that support workers into the future. Ongoing monitoring of this cohort into 2022 will provide further insights into the pandemic’s longitudinal impacts on HCWs and help identify organisational responses that best support their wellbeing.

## Figures and Tables

**Figure 1 ijerph-19-04951-f001:**
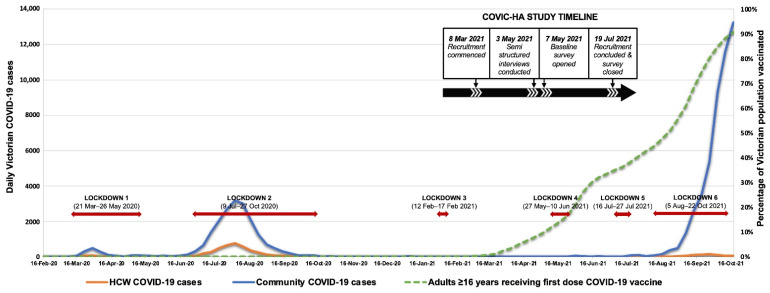
COVID-19 case numbers sourced from Victorian COVID-19 data available at https://www.coronavirus.vic.gov.au/victorian-coronavirus-covid-19-data (accessed on 18 November 2021); Victorian healthcare worker (clinical and non-clinical) COVID-19 data available at https://www.coronavirus.vic.gov.au/healthcare-worker-covid-19-data (accessed on 18 November 2021). COVID-19 vaccine data for Victoria were sourced from https://www.coronavirus.vic.gov.au/weekly-covid-19-vaccine-data#covid-19-vaccine-rates-for-second-dose-by-postcode (accessed on 18 November 2021).

**Table 1 ijerph-19-04951-t001:** Baseline characteristics of COVIC-HA study population by occupational groups.

Characteristics	Total Cohort n (%)	Occupational Group				
		Paramedics n (%)	Nurses n (%)	Allied Health ^ n (%)	Doctors n (%)	Others * n (%)
**Overall**	**984 (100.0%)**	**126 (12.8%)**	**319 (32.4%)**	**174 (17.7%)**	**123 (12.5%)**	**242 (24.6%)**
**Gender**						
Male	267 (27.1%)	70 (55.6%)	37 (11.6%)	28 (16.1%)	66 (53.7%)	66 (27.3%)
Female	714 (72.6%)	55 (43.7%)	282 (88.4%)	145 (83.3%)	57 (46.3%)	175 (72.3%)
Non-binary	3 (0.3%)	1 (0.8%)	0 (0.0%)	1 (0.6%)	0 (0.0%)	1 (0.4%)
**Age**						
<45 years	499 (50.7%)	83 (65.9%)	151 (47.3%)	98 (56.3%)	62 (50.4%)	105 (43.4%)
≥45 years	485 (49.3%)	43 (34.1%)	168 (52.7%)	76 (43.7%)	61 (49.6%)	137 (56.6%)
**Work setting**						
Hospital	808 (82.1%)	0 (0.0%)	309 (96.9%)	168 (96.6%)	112 (91.1%)	219 (90.5%)
Ambulance	141 (14.3%)	126 (100.0%)	2 (0.6%)	2 (1.1%)	0 (0.0%)	11 (4.5%)
Aged care	7 (0.7%)	0 (0.0%)	1 (0.3%)	3 (1.7%)	0 (0.0%)	3 (1.2%)
Primary care	28 (2.8%)	0 (0.0%)	7 (2.2%)	1 (0.6%)	11 (8.9%)	9 (3.7%)
**COVID Infection Status** (n = 983)						
COVID-19 Infected	22 (2.2%)	2 (1.6%)	11 (3.4%)	1 (0.6%)	1 (0.8%)	7 (2.9%)
Furloughed but not infected	203 (20.7%)	47 (37.3%)	79 (24.8%)	33 (19.0%)	15 (12.2%)	29 (12.0%)
Neither infected nor furloughed	758 (77.1%)	77 (61.1%)	229 (71.8%)	140 (80.5%)	107 (87.0%)	205 (85.1%)
**Work Experience**						
≤10 years	386 (39.2%)	66 (52.4%)	101 (31.7%)	64 (36.8%)	32 (26.0%)	123 (50.8%)
>10 years	598 (60.8%)	60 (47.6%)	218 (68.3%)	110 (63.2%)	91 (74.0%)	119 (49.2%)

^ Includes audiology, chiropractic, dietetics, exercise physiology, medial radiation, OT, optometry, orthotics/prosthetics, pharmacy, physiotherapy, psychology, social work, speech pathology. * Includes administrative and clerical staff, orderlies, food service staff, information technology, engineering etc.

**Table 2 ijerph-19-04951-t002:** Measures of depression, anxiety, post-traumatic stress, resilience, wellbeing, life satisfaction, burnout, and optimism across occupations.

Validated Scales	Total Cohort n (%)	Paramedics n (%)	Nurses n (%)	Allied Health n (%)	Doctors n (%)	Others n (%)
**Overall**	**984 (100%)**	**126 (12.8%)**	**319 (32.4%)**	**174 (17.7%)**	**123 (12.5%)**	**242 (24.6%)**
**Patient Health Questionnaire-9 (PHQ-9)—Depression**						
Mean (SD)	6.0 (5.4)	7.3 (5.5)	6.9 (5.8)	5.0 (4.8)	4.3 (4.1)	6.0 (5.4)
≥10 (Moderate-Severe)	221 (22.5%)	40 (31.7%)	81 (25.4%)	27 (15.5%)	15 (12.2%)	58 (24.0%)
<10 (Minimal-Mild)	763 (77.5%)	86 (68.3%)	238 (74.6%)	147 (84.5%)	108 (87.8%)	184 (76.0%)
**PHQ item 9 (thoughts of being better off dead or self-harm)**						
Several days or more frequently	87 (8.8%)	12 (9.5%)	39 (12.2%)	10 (5.7%)	7 (5.7%)	19 (7.9%)
Not at all	897 (91.2%)	114 (90.5%)	280 (87.8%)	164 (94.3%)	116 (94.38%)	223 (92.1%)
**Generalised Anxiety Disorder 7-item scale (GAD-7)—Anxiety**						
Mean (SD)	4.6 (4.7)	5.1 (4.6)	5.1 (4.9)	3.9 (4.5)	3.2 (3.7)	4.8 (4.8)
≥10 (Moderate-Severe)	138 (14.0%)	21 (16.7%)	54 (16.9%)	15 (8.6%)	10 (8.1%)	38 (15.7%)
<10 (Minimal-Mild)	846 (86.0%)	105 (83.3%)	265 (83.1%)	159 (91.4%)	113 (91.9%)	204 (84.3%)
**GAD-7 item 6 (becoming easily annoyed or irritable)**						
Several days or more frequently	599 (60.9%)	86 (68.3%)	210 (65.82%)	89(51.1%)	71 (57.7%)	143 (59.1%)
Not at all	385 (39.1%)	40 (31.7%)	109 (34.2%)	85 (48.9%)	52 (42.3%)	99 (40.9%)
**Impact of Event Scale-6 (IES-6)—Post-traumatic stress disorder (PTSD)**						
Mean (SD) *	0.9 (0.8)	1.1 (0.9)	1.0 (0.9)	0.8 (0.7)	0.7 (0.7)	1.0 (0.9)
>9 (Moderate-Severe) ^#^	201 (20.4%)	32 (25.4%)	80 (25.1%)	21 (12.1%)	12 (9.8%)	56 (23.1%)
≤9 (None/Minimal) ^#^	783 (79.6%)	94 (74.6%)	239 (74.9%)	153 (87.9%)	111 (90.2%)	186 (76.9%)
**Connor-Davidson Resilience Scale 2-item (CD-RISC-2)—Resilience**						
Mean (SD)	6.3 (1.4)	6.3 (1.4)	6.2 (1.3)	6.3 (1.4)	6.4 (1.4)	6.3 (1.3)
**Personal wellbeing index-Adult (PWI-A)—Wellbeing**						
PWI-A (Well-being)—Mean (SD)	69.7 (17.3)	67.5 (17.3)	67.6 (18.4)	72.8 (14.6)	76.2 (13.9)	68.3 (17.9)
PWI Item 1 (Life Satisfaction)—Mean (SD)	68.9 (18.9)	66.8 (19.0)	66.5 (20.4)	71.3 (15.8)	75.1 (14.9)	68.1 (19.9)
**Abbreviated Maslach Burnout Inventory (aMBI)—Burnout**						
**Emotional Exhaustion (n = 941)**						
Mean (SD)	8.7 (4.7)	9.5 (4.9)	9.6 (4.7)	8.4 (4.5)	7.7 (4.2)	7.8 (5.0)
≥7 (Moderate-Severe burnout)	613 (65.1%)	85 (68.5%)	223 (71.7%)	109 (64.5%)	74 (60.7%)	122 (56.7%)
<7 (No-Low burnout)	328 (34.9%)	39 (31.5%)	88 (28.3%)	60 (35.5%)	48 (39.3%)	93 (43.3%)
**Depersonalization (n = 803)**						
Mean (SD)	3.0 (3.9)	5.9 (4.8)	2.9 (3.7)	2.2 (3.0)	2.5 (3.2)	2.1 (3.6)
≥4 (Moderate-Severe burnout)	252 (31.4%)	71 (60.7%)	95 (32.1%)	36 (22.9%)	29 (24.4%)	21 (18.4%)
<4 (No-Low burnout)	551 (68.6%)	46 (39.3%)	201 (67.9%)	121 (77.1%)	90 (75.6%)	93 (81.6%)
**Personal Accomplishment (n = 772)**						
Mean (SD)	14.3 (2.9)	13.2 (3.3)	14.4 (2.9)	14.7 (2.6)	14.6 (2.8)	14.3 (3.0)
≤14 (Moderate-Severe burnout)	348 (45.1%)	74 (61.2%)	133 (45.9%)	55 (36.9%)	45 (38.1%)	41 (43.6%)
>14 (No-Low burnout)	424 (54.9%)	47 (38.8%)	157 (54.1%)	94 (63.1%)	73 (61.9%)	53 (56.4%)
**Optimism**						
Mean (SD)	7.0 (1.9)	6.9 (2.0)	6.9 (1.9)	7.2 (1.6)	7.3 (1.9)	7.1 (1.9)
<8 Low-moderate optimism)	655 (66.6%)	86 (68.3%)	225 (70.5%)	114 (65.5%)	68 (55.3%)	162 (66.9%)
≥8 High optimism	329 (33.4%)	40 (31.7%)	94 (29.5%)	60 (34.5%)	55 (44.7%)	80 (33.1%)

* Mean across all six IES items; ^#^ sum of responses to each IES item.

**Table 3 ijerph-19-04951-t003:** Comparison * of continuous and binary outcomes for measures of depression, anxiety, post-traumatic stress, resilience, wellbeing, life satisfaction, burnout, and optimism across occupations.

Validated Scales	Nurses vs. Paramedics Mean Difference (95% CI)	Allied Health vs. Paramedics Mean Difference (95% CI)	Doctors vs. Paramedics Mean Difference (95% CI)	Others vs. Paramedics Mean Difference (95% CI)	*p*-Value	Nurses vs. Paramedics Mean Difference (95% CI)	Allied Health vs. Paramedics Mean Difference (95% CI)	Doctors vs. Paramedics Mean Difference (95% CI)	Others vs. Paramedics Mean Difference (95% CI)	*p*-Value **
	**Comparison of Continuous Outcomes**					**Comparison of Binary Outcomes**				
**PHQ-9—Depression**	0.5 (−0.7, 1.7)	−1.3 (−2.6, 0.0)	−1.8 (−3.3, −0.4)	−0.3 (−1.5, 0.9)	<0.001	1.16 (0.81, 1.65)	0.71 (0.45, 1.12)	0.57 (0.33, 1.00)	1.05 (0.73, 1.51)	0.024
**GAD-7—Anxiety**	0.8 (−0.3, 1.8)	−0.4 (−1.5, 0.7)	−0.8 (−2.1, 0.4)	0.6 (−0.5, 1.6)	0.006	1.40 (0.85, 2.30)	0.72 (0.37, 1.39)	0.76 (0.36, 1.60)	1.30 (0.78, 2.18)	0.070
**IES-6 Post-traumatic stress**	0.1 (−0.2, 0.3)	−0.1 (−0.4, 0.1)	−0.2 (−0.4, 0.0)	0.1 (−0.2, 0.3)	0.004	1.15 (0.78, 1.70)	0.55 (0.33, 0.93)	0.47 (0.25, 0.88)	1.08 (0.72, 1.61)	0.001
**CD-RISC-2—Resilience**	−0.1 (−0.8, 0.5)	−0.0 (−0.7, 0.6)	0.1 (−0.6, 0.8)	−0.0 (−0.7, 0.6)	0.65	N/A	N/A	N/A	N/A	N/A
**PWI-A—Wellbeing**										
PWI-A (Well-being)	−2.1 (−7.4, 3.3)	2.6 (−2.9, 8.2)	6.9 (1.3, 12.6)	−0.9 (−6.2, 4.3)	<0.001	N/A	N/A	N/A	N/A	N/A
PWI Item 1 (Life Satisfaction)	−1.7 (−9.2, 5.9)	2.7 (−4.9, 10.3)	7.4 (−0.3, 15.1)	0.1 (−7.2, 7.4)	<0.001	N/A	N/A	N/A	N/A	N/A
**aMBI—Burnout**										
Emotional Exhaustion	0.6 (−1.2, 2.3)	−0.7 (−2.3, 0.9)	−1.2 (−2.8, 0.5)	−1.1 (−2.6, 0.5)	<0.001	1.14 (0.99, 1.32)	0.99 (0.84, 1.17)	0.96 (0.80, 1.15)	0.90 (0.76, 1.06)	0.002
Depersonalization	−2.3 (−3.5, −1.1)	−3.2 (−4.4, −1.9)	−3.1 (−4.3, −1.8)	−3.1 (−4.3, −1.9)	<0.001	0.56 (0.31, 1.04)	0.39 (0.22, 0.69)	0.43 (0.24, 0.77)	0.34 (0.19, 0.60)	0.003 ^a^
Personal Accomplishment	0.9 (−0.3, 2.0)	1.4 (0.3, 2.5)	1.3 (0.2, 2.4)	0.8 (−0.2, 1.9)	0.017	0.86 (0.52, 1.42)	0.67 (0.41, 1.10)	0.67 (0.41, 1.09)	0.81 (0.52, 1.26)	0.15 ^b^
**Optimism**	−2.5 (−9.2, 4.3)	1.2 (−5.5, 8.0)	4.0 (−2.7, 10.8)	0.6 (−5.9, 7.1)	0.029	1.06 (0.82, 1.36)	0.97 (0.78, 1.22)	0.83 (0.65, 1.05)	1.01 (0.81, 1.26)	0.16 ^a^

* The models used to estimate differences between means or relative risks are adjusted for SES decile, age, an indicator of date of participation (pre/post fourth lockdown), and gender (female vs. not female) unless otherwise indicated. Relative risks correspond to the risk of having the poorer psychological outcome category (i.e., the categories listed first in Table 2). ** *p*-values correspond to the test of the global null hypothesis of no difference between occupations. ^a^ Adjusted for age. ^b^ Adjusted for age, gender, and date of participation.

**Table 4 ijerph-19-04951-t004:** HCW experiences of the pandemic—dominant themes, sub-themes, and illustrative quotes from analysis of free-text responses.

Themes and Subthemes ^#^	Illustrative Quote (Occupational Group, Healthcare Setting)
**Working in a complex and changing environment** Physical and emotional toll Keeping up with frequent policy and practice changes Moral injury, especially related to visitor restrictions Inescapability, difficulty “switching off”	“Working through COVID in healthcare has been incredibly stressful and draining, we’re nowhere near finished, and most of us are trying to pour from empty cups.” (Allied health worker, hospital) “In a role that is very time poor, it was and remains a constant battle to ensure we are up to date with the latest protocols, knowledge, risks and assessments about Covid.” (Nurse, hospital) “I have never worked so hard in my entire career. I feel mentally and physically drained after work and it takes all of my days off to recuperate then I am back at work being brought down again.” (Paramedic, ambulance)
**Concerns about patient/resident care** Changed patient-clinician interface Isolation and deconditioning Long term impacts	“The impact of isolation on residents was heartbreaking and resulted in significant deconditioning which we are only seeing now.” (Nurse, hospital) “As a social worker we are working with the shadow pandemic of family violence and mental illness to a degree we have never seen before. This will go on for years to come–trying to pick up the pieces.” (Allied health, hospital)
**Isolation and disconnection** Fear of bringing COVID-19 home Personal and profession isolation	“My fear of passing something on, as well as the sadness of limiting my contact [with family] out of those fears has had a significant negative impact on my general wellbeing and family closeness.” (Paramedic, ambulance) “I found the whole experience frightening—I can only speak for myself but I was living in fear every day of getting COVID or giving it to someone.” (Nurse, hospital)
**Positive experiences** Rising to the challenge Renewed perspective	“I did feel a sense of pride about the way our team rose to the challenge and provided such empathetic and respectful care in such a challenging time.” (Nurse, hospital) “The experience with Covid-19 made me perceive the world differently. I no longer want to keep working extra hours and decided to work 4 days a week. I decided to spend time on myself and surrounding people more than pre-covid.” (Nurse, aged-care)

^#^ Themes appear in bold text; sub themes appear as bullet points.

## Data Availability

The data presented in this study are available upon request from the corresponding author. The data are not publicly available due to the ongoing nature of the COVIC-HA project.

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
