# Peer review of "Mental Health Outcomes in Australian Healthcare and Aged-Care Workers during the Second Year of the COVID-19 Pandemic"

_ijerph, 2022, doi:10.3390/ijerph19094951_

Round 1
Reviewer 1 Report
Good paper, well worth publishing after minor touch ups (see comments in attached report).

Author Response
Response to reviewers
We would like to thank the reviewers for their thoughtful and constructive comments. Here we provide a point-by-point response to all issues raised. Reviewer comments and questions appear in bold text, and author responses in plain text.
Reviewer: 1
This is a good, well-written paper, by authors who are clearly experienced in this type of exercise. A solid introduction, an exhaustive review of the literature like we rarely see, a more than adequate methodology clearly described, state-of-the-art figures and tables (I found Figure 1 particularly impressive), an excellent standard of language, thorough analyses, a relevant discussion, this paper has all the attributes of a publishable article. However, a few minor points would deserve attention prior to publication.
For instance, lines 235 to 237 seem to be footnotes and should be presented as such; if that is not the case, then they should be better linked into the rest of the text.
Author response: You are correct - these lines are footnotes to Figure 1. We have corrected this formatting error.
Secondly, in the abstract the authors write: “We established the Coronavirus in Victorian Healthcare and Aged-Care Workers”; this sentence in the abstract reads a bit strange; it sounds like the researchers contaminated the workers. Given that “established” most probably applies to “cohort” (quite a bit further in the sentence), I suggest to remove the words “Coronavirus in”.
Author response: Our intention was to introduce our study by name (i.e. the “Coronavirus in Victorian Healthcare and Aged Care Workers (COVIC-HA)” study), but we can see how this sentence might be misinterpreted. We have changed this sentence (page 1, line 19-20) as follows: “We established a Victorian HCW cohort (the Coronavirus in Victorian Healthcare and Aged Care Workers [COVIC-HA] study) to examine...”
Third, the authors mention: “Our study is the only Australian and one of few reported studies assessing COVID-210 19 impacts on HCWs in 2021”. They do a detailed job of comparing their results (Tables 3 to 6 incl.) with just about everything that has been published on the subject. The authors should highlight a bit more clearly their specific, original contribution. Finally, whereas the authors are fully conscious of the limitations of their research (lines 274-292), it nevertheless remains that they have available an incredible wealth of good data. They performed quite exhaustive analyses of these data, providing an impressive amount of results. Given all these results, I found the conclusion a bit shy. One would expect the authors to propose, even if tentatively, solutions to help HCWs prevent or cope with “clinically significant depression, anxiety and PTS symptoms”. Additional monitoring into 2022 is OK, a significant amount of literature has measured the extent of the problem, but prevention has become a concern by now. Otherwise this is a fine paper, well worth publishing after minor touch-ups.
Author response: We thank the reviewer for this comment. In response, we have added the following sentence to the end of our limitations paragraph (page 11, line 320-323):
“Nevertheless, our data provide an indication of the mental health impact of COVID-19 on HCWs across a range of occupations in the pandemic’s second year, and ongoing monitoring of this longitudinal cohort will enable us to monitor trends over time.”
We have also added a sentence to the conclusions paragraph (page 11, line 332-335) as follows:
“Given the ongoing risk of COVID-19 and evidence of significant impacts on HCWs, it is essential that policymakers and healthcare organisations instigate sustainable mitigation strategies and targeted interventions that support workers into the future.”
Reviewer 2 Report
The present paper aimed at investigating the impact of COVID-19 on mental health symptoms and on wellbeing and personal functioning of Australian healthcare and aged-care workers in the May-July 2021 period. Nine hundred eighty-four subjects completed an online survey including personal characteristics, pandemic experiences and psychiatric assessment. The results of the present cross-sectional online survey showed as nurses and paramedics reported high depressive, anxiety, post-traumatic stress and burnout symptoms.
Comments:
1) Introduction, Lines 39-50. The literature review can be improved for this background, and more international evidence should be provided (see also point 3).
2) Furthermore, in the Introduction section, more detailed information about the impact of COVID-19 on the Australian population and healthcare workers/system during the present survey should be provided. Authors reported, indeed, that there was a strong reduction of number of COVID-19 cases during their survey. In my opinion, I think it could be useful to move the Figure 1 from Discussion to the Introduction section and describe/comment it.
3) As reported by Author, few studies examined differences in COVID-19-related experiences across healthcare settings. As reported in Methods, subjects were recruited from four different healthcare settings (hospital, ambulance, aged-care, primary care). There were any differences in the impact of COVID-19 on the study variables across the four different healthcare settings? It would be of interest to Readers if the Authors provided data on this point, comparing their results with those currently available in the international literature (e.g. 1177/21501327211039714; 10.1016/j.jad.2021.10.128; 10.1136/bmjopen-2020-042030; etc…).
4) Were there any issues of Common Method Variance bias?
5) The use of self-report instruments that could be considered less accurate than a clinician assessment, besides the cross-sectional design and the use of a convenient sample (with an online survey) represents important limitations that should be properly reported in the Limitations’ paragraph.
Author Response
Response to reviewers
We would like to thank the reviewers for their thoughtful and constructive comments. Here we provide a point-by-point response to all issues raised. Reviewer comments and questions appear in bold text, and author responses in plain text.
Reviewer 2:
The present paper aimed at investigating the impact of COVID-19 on mental health symptoms and on wellbeing and personal functioning of Australian healthcare and aged-care workers in the May-July 2021 period. Nine hundred eighty-four subjects completed an online survey including personal characteristics, pandemic experiences and psychiatric assessment. The results of the present cross-sectional online survey showed as nurses and paramedics reported high depressive, anxiety, post-traumatic stress and burnout symptoms.
Comments:
1) Introduction, Lines 39-50. The literature review can be improved for this background, and more international evidence should be provided (see also point 3).
Author response: We have added additional literature to the introduction, including international evidence, in response to this suggestion. The updated text (page 2, line 57-62) now reads:
“International experience has shown healthcare workers (HCWs) to be at high risk of both COVID-19 infection [8] and significant psychosocial impacts from the pandemic, including post-traumatic stress disorder (PTSD), anxiety, burnout, and depression [9-13]. Reports also indicate occupational disparities in COVID-19 infection risks, with HCWs facing greater risks than the general community, especially those in frontline roles. [8,1415]”
2) Furthermore, in the Introduction section, more detailed information about the impact of COVID-19 on the Australian population and healthcare workers/system during the present survey should be provided. Authors reported, indeed, that there was a strong reduction of number of COVID-19 cases during their survey. In my opinion, I think it could be useful to move the Figure 1 from Discussion to the Introduction section and describe/comment it.
Author response: We have moved Figure 1 to the introduction (page 2) and provided additional contextual information on the pandemic as experienced in Australia (with a focus on the State of Victoria) in response to this suggestion. The additional text (page 1-2, line 44-57) reads:
“Victoria experienced the greatest burden of COVID-19 in Australia in 2020, although the overall attack rate of 3,023 per million at the end of 2020 was far less than the US (56,341/million) or the UK (33,232/million) [3, 4]. An initial wave of infection (Jan-April 2020) was contained through public health interventions and a second wave (May-Nov 2020) was eventually suppressed with aggressive restrictions, including prolonged lockdowns, leading to local elimination (Figure 1). Subsequent small outbreaks were swiftly suppressed via contact tracing of identified cases and short periods of increased restrictions (including lockdowns), with case numbers remaining very low until incursion of the SARS-CoV-2 Delta variant in June 2021 led to escalating case numbers (despite further lockdowns) [4, 5]. A shift in strategy then occurred away from elimination and towards endemicity plus high levels of vaccination [5]. The psychological burden of the pandemic upon the Australian general population has been high, and extended lockdowns in Victoria have led to increased psychological distress [6, 7].”
3) As reported by Author, few studies examined differences in COVID-19-related experiences across healthcare settings. As reported in Methods, subjects were recruited from four different healthcare settings (hospital, ambulance, aged-care, primary care). Were there any differences in the impact of COVID-19 on the study variables across the four different healthcare settings? It would be of interest to Readers if the Authors provided data on this point, comparing their results with those currently available in the international literature (e.g. 1177/21501327211039714; 10.1016/j.jad.2021.10.128; 10.1136/bmjopen-2020-042030; etc…).
Author response: We chose not to include data on comparisons across healthcare settings in our manuscript as the small sample sizes of participants from aged care and primary care precluded meaningful analyses across settings. Comparisons between hospital and ambulance settings (excluding aged care and primary care) are unlikely to offer much new information as most ambulance staff were paramedics (Table 1), and comparisons between hospital and ambulance settings are thus akin to comparisons between paramedics and the combined group of nurses/doctors/allied health/other. We have added the following sentence to the limitations section (page 10, line 299-300):
Small numbers of participants in aged care and primary care settings precluded meaningful analyses across healthcare settings.
4) Were there any issues of Common Method Variance bias?
5) The use of self-report instruments that could be considered less accurate than a clinician assessment, besides the cross-sectional design and the use of a convenient sample (with an online survey) represents important limitations that should be properly reported in the Limitations’ paragraph.
Author response: We understand common method variance bias in this study context to refer to artificial inflation of correlation between outcomes due to factors such as being self-reported and potentially using similar scales. As our primary psychological outcomes used a variety of validated scales and we analysed each of the outcome measures separately, we do not expect our results to have been impacted by common method variance bias. We acknowledge that the use of self-reported measures and use of a convenience sample are limitations of our approach and have updated our limitations section accordingly. The updated section (page 10-11, line 300-308) reads as follows:
“although we used validated instruments to measure mental health outcomes [20-25], self-reported measures may not directly correlate with clinical diagnostic interviews[44]. However, as our primary outcome measures used validated scales and we analysed each outcome measure separately, we do not expect our results to have been impacted by common method variance bias. Third, convenience sampling, voluntary participation and low response rates likely resulted in response biases. Attempts to minimise bias included addressing potential confounders by adjusting for age, gender, socio-economic status and date of participation in analyses.